# ViCA-NeRF: View-Consistency-Aware 3D Editing of Neural Radiance Fields

**Jiahua Dong**      **Yu-Xiong Wang**
University of Illinois Urbana-Champaign
{jiahuad2, yxw}@illinois.edu

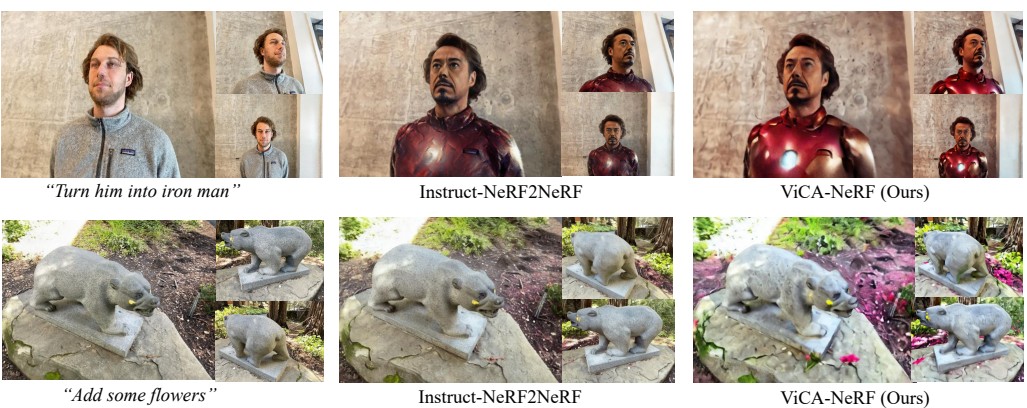

*"Turn him into iron man"*          Instruct-NeRF2NeRF          ViCA-NeRF (Ours)

*"Add some flowers"*          Instruct-NeRF2NeRF          ViCA-NeRF (Ours)

Figure 1: Our ViCA-NeRF is the first work that achieves **multi-view consistent 3D editing** with text instructions, applicable across a broad range of scenes and instructions. Moreover, ViCA-NeRF exhibits controllability, allowing for early control of final results by editing key views. Notably, ViCA-NeRF is also efficient, surpassing state-of-the-art Instruct-NeRF2NeRF by being 3 times faster.

## Abstract

We introduce ViCA-NeRF, the *first* view-consistency-aware method for 3D editing with text instructions. In addition to the implicit neural radiance field (NeRF) modeling, our key insight is to exploit two sources of regularization that *explicitly* propagate the editing information across different views, thus ensuring multi-view consistency. For *geometric regularization*, we leverage the depth information derived from NeRF to establish image correspondences between different views. For *learned regularization*, we align the latent codes in the 2D diffusion model between edited and unedited images, enabling us to edit key views and propagate the update throughout the entire scene. Incorporating these two strategies, our ViCA-NeRF operates in two stages. In the initial stage, we blend edits from different views to create a preliminary 3D edit. This is followed by a second stage of NeRF training, dedicated to further refining the scene's appearance. Experimental results demonstrate that ViCA-NeRF provides more flexible, efficient (3 times faster) editing with higher levels of consistency and details, compared with the state of the art. Our code is available at: https://github.com/Dongjiahua/VICA-NeRF.

## 1   Introduction

The recent advancements in 3D reconstruction technology, exemplified by the neural radiance field (NeRF) [1] and its variants, have significantly improved the convenience of collecting real-world

37th Conference on Neural Information Processing Systems (NeurIPS 2023).

3D data. Such progress has opened up new opportunities for the development of various 3D-based applications. By capturing RGB data from multiple views and obtaining corresponding camera parameters, NeRF enables efficient 3D representation and rendering from any viewpoint. As the availability of diverse 3D data increases, so does the demand for *3D editing* capabilities.

However, performing 3D editing on NeRF is not straightforward. The implicit representation of NeRF makes it challenging to directly modify the 3D scene. On the other hand, since NeRF is trained on RGB images, it can effectively leverage a wealth of existing 2D models. Motivated by this, state-of-the-art methods, such as Instruct-NeRF2NeRF [2], apply 2D editing on individual images via a text-instructed diffusion model (e.g., Instruct-Pix2Pix [3]), and then rely *exclusively* on NeRF to learn from the updated dataset with edited images and propagate the editing information across different views. Despite yielding encouraging results, this *indirect* 3D editing strategy is time-consuming, because of its iterative process that requires cyclically updating the image dataset while progressively fine-tuning NeRF parameters. And more seriously, due to the lack of inherent 3D structure, the editing results from the 2D diffusion model often exhibit significant variations across different views. This raises a critical concern regarding 3D inconsistency when employing such a strategy, particularly in cases where the editing operation is aggressive, e.g., *"Add some flowers"* and *"Turn him into iron man"* as illustrated in Figure 1.

To overcome these limitations, in this work we introduce ViCA-NeRF, a View-Consistency-Aware NeRF editing method. Our *key insight* is to exploit *two additional sources of regularization* that *explicitly* connect the editing status across different views and correspondingly propagate the editing information from edited to unedited views, thus ensuring their multi-view consistency. The first one is *geometric regularization*. Given that depth information can be obtained from NeRF for free, we leverage it as a guidance to establish image correspondences between different views and then project edited pixels in the edited image to the corresponding pixels in other views, thereby enhancing consistency. This approach allows us to complete preliminary data modifications before fine-tuning NeRF, as well as enabling users to determine 3D content by selecting the editing results from key 2D views. Moreover, as our method allows for direct and consistent edits, it is more efficient by avoiding the need for iterative use of the diffusion model.

The second one is *learned regularization*. The depth information derived from NeRF can often be noisy, yielding certain incorrect correspondences. To this end, we further align the latent codes in the 2D diffusion model (e.g., Instruct-Pix2Pix) between edited and unedited images via a blending refinement model. Doing so updates the dataset with more view-consistent and homogeneous images, thereby facilitating NeRF optimization.

Our ViCA-NeRF framework then operates in two stages, with these two crucial regularization strategies integrated in the first stage. Here, we edit key views and blend such information into every viewpoint. To achieve this, we employ the diffusion model for blending after a simple projection mixup. Afterward, the edited images can be used directly for training. An efficient warm-up strategy is also proposed to adjust the editing scale. In the second stage, which is the NeRF training phase, we further adjust the dataset. Our method is evaluated on various scenes and text prompts, ranging from faces to outdoor scenes. We also ablate different components in our method and compare them with previous work [2]. Experimental results show that ViCA-NeRF achieves higher levels of consistency and detail, compared with the state of the art.

Our **contributions** are three-fold. (1) We propose ViCA-NeRF, the *first* work that explicitly enforces multi-view consistency in 3D editing tasks. (2) We introduce geometric and learned regularization, making 3D editing more flexible and controllable by editing key views. (3) Our method significantly improves the efficiency of 3D editing, achieving a speed 3 times faster than the state of the art.

## 2   Related Work

**Text-to-Image Diffusion Models for 2D Editing.** Diffusion models have recently demonstrated significant capabilities in text-to-image editing tasks [3–8]. Earlier studies have leveraged pre-trained text-to-image diffusion models for image editing. For instance, SDEdit [8] introduces noise into an input image and subsequently denoises it through a diffusion model. Despite achieving reasonable results, this approach tends to lose some of the original image information. Other studies have concentrated on local inpainting using provided masks [4, 5], enabling the generation of corresponding content within the masked area with text guidance. More recently, Instruct-Pix2Pix [3] proposes an

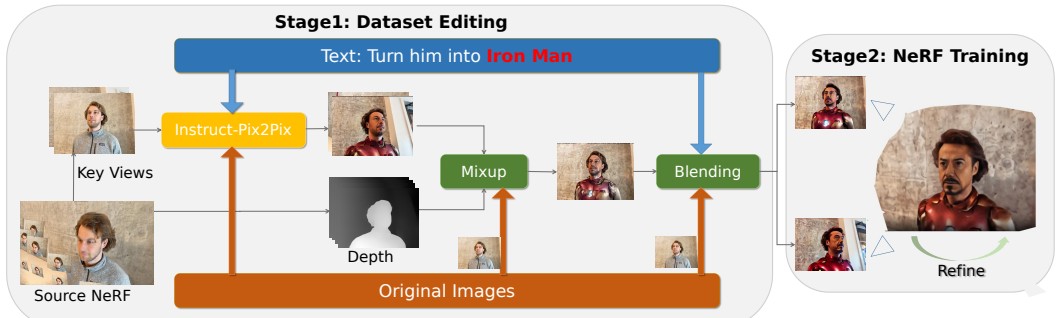

Figure 2: **Overview of our ViCA-NeRF.** Our proposed method decouples NeRF editing into two stages. In the first stage, we sample several key views and edit them through Instruct-Pix2Pix. Then, we use the depth map and camera poses to project edited keyframes to other views and obtain a mixup dataset. These images are further refined through our blending model. In the second stage, the edited dataset is directly used to train the NeRF model. Optionally, we can conduct refinement to the dataset according to the updated NeRF.

approach to editing images based on text instructions. Instruct-Pix2Pix accepts original images and text prompts as conditions. Trained on datasets of editing instructions and image descriptions, this approach can effectively capture consistent information from the original image while adhering to detailed instructions.

In this paper, we adopt Instruct-Pix2Pix as our editing model. Importantly, we further explore the potential of Instruct-Pix2Pix by extending its application to editing, blending, and refinement tasks in more challenging 3D editing scenarios.

**Implicit 3D Representation.** Recently, neural radiance field (NeRF) and its variants have demonstrated significant potential in the field of 3D modeling [1, 9–14]. Unlike explicit 3D representations such as point clouds or voxels, NeRF uses an implicit representation to capture highly detailed scene geometry and appearance. While training a conventional NeRF model can be time-consuming, Instant-NGP [10] accelerates this process by training a multi-layer perceptron (MLP) with multi-resolution hash input encoding. Notably, NeRFStudio [9] offers a comprehensive framework that can handle various types of data. It modularizes each component of NeRF, thereby supporting a broad range of data. These efforts bridge the gap between image data and 3D representation, making it possible to guide 3D generation from 2D models. Our work uses the 'nerfacto' model from NeRFStudio to fit real scenes. We also leverage the depth estimated from the model to maintain consistency.

**3D Generation.** Inspired by the success of 2D diffusion models [15, 16], 3D generation has drawn great attention in recent years [17–21]. Dream Fusion first proposes to use a score distillation sampling (SDS) loss as the training target. Later, Magic3D [20] generates higher resolution results by a coarse-to-fine procedure. Various scenarios have been explored, including generating a detailed mesh from text prompts [21] and generating novel objects based on images from a few views [18].

Rather than generating new content, our work focuses on editing the 3D representation with text prompts. By explicitly leveraging 2D editing, we provide a more direct approach to control the final 3D result.

**NeRF Editing.** Because of the implicit representation of NeRF [1], editing NeRF remains a challenge. Early attempts [22–32] perform simple editing on shape and color, by using the guidance from segmentation masks, text prompts, etc. NeuralEditor [33] proposes to exploit a point cloud for editing and obtain the corresponding NeRF representation. Another line of work tries to modify the content through text prompts directly. NeRF-Art [34] uses a pre-trained CLIP [35] model to serve as an additional stylized target for the NeRF model. More recently, Instruct-NeRF2NeRF [2] and Instrcut-3D-to-3D [36] leverage Instruct-Pix2Pix [3] for editing. They can edit the content of NeRF with language instructions. However, both require using the diffusion model iteratively in the NeRF training process, making them less efficient. More importantly, though it shows promising results on real-world data, Instruct-NeRF2NeRF can only work when the 2D diffusion model produces consistent results in 3D. Thus, it is unable to generate a detailed result where 2D edits tend to be different, and it also suffers from making diverse results from the same text prompt.

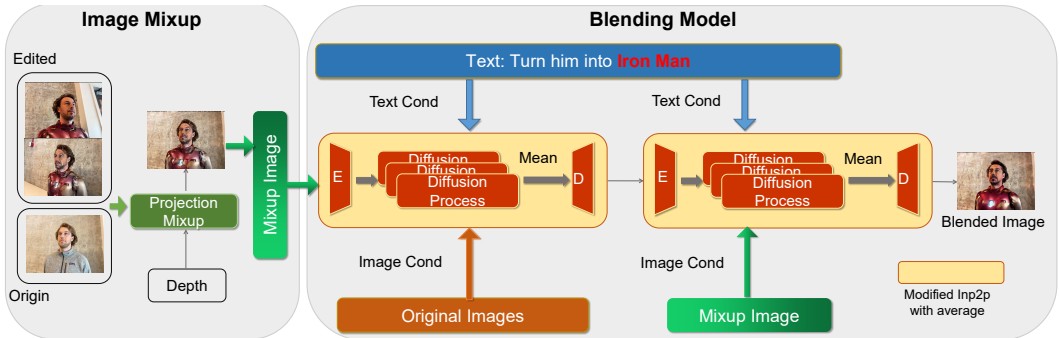

Figure 3: **Illustration of mixup procedure and blending model.** We first mix up the image with the edited key views. Then, we introduce a blending model to further refine it. The blending model utilizes two modified Instruct-Pix2Pix ('Inp2p') processes. In each process, we generate multiple results and take their average on the latent code to decode a single final result.

In our work, we propose to use depth guidance for detailed 3D editing. Our ViCA-NeRF only needs to update the dataset at the start of training and further refine it once, making it significantly more efficient. In addition, our method results in more detailed and diverse edit results compared with Instruct-NeRF2NeRF. As several key views guide the whole 3D scene, we allow users to personalize the final effect on NeRF by choosing 2D edit results.

## 3 Method

Our ViCA-NeRF framework is shown in Figure 2. The main idea is to use key views to guide the editing of other views, ensuring better consistency and detail. Given the source NeRF, we first extract each view's depth from it and sample a set of key views. Then, we edit these key views using Instruct-Pix2Pix [3] and propagate the edit result to other views. To address the problem of incorrect depth estimation from NeRF, we perform mixup on each image and then refine it through our blending model. The obtained data can be used directly for NeRF training. Optionally, we can perform post-refinement to the dataset after obtaining more consistent rendering results from NeRF.

### 3.1 Preliminary

We first elaborate on NeRF, including how we obtain rendered RGB images and coarse depth from the NeRF model. Subsequently, we present Instruct-NeRF2NeRF, a diffusion model for 2D editing. With this model, we can perform 2D editing using text and then further enable 3D editing.

**Neural Radiance Field.** NeRF [1] and its variants [9, 10] are proposed for 3D scene representation. The approach models a 3D scene as a 5D continuous function, mapping spatial coordinate $(x, y, z)$ and viewing direction $(\theta, \phi)$ to volume density $\sigma$ and emitted radiance $c$. For a given set of 2D images, NeRF is trained to reproduce the observed images. This is achieved by volume rendering, where rays are traced through the scene and colors are accumulated along each ray to form the final rendered image. While NeRF does not explicitly predict depth, one common approach is to compute a weighted average of the distance values along each ray, where the weights are the predicted volume densities. This process can be conceptualized as approximating the depth along the ray.

**Instruct-Pix2Pix.** Instruct-Pix2Pix [3] is a 2D diffusion model for image editing. It takes a text prompt $c_T$ and a guidance image $c_I$ as conditions. Given an image $z_0$, it first adds noise to the image to create $z_t$, where $t$ is the timestep to start denoising. With denoising U-Net $\epsilon_\theta$, the predicted noise can be calculated as

$$\widetilde{\epsilon} = \epsilon_\theta(z_t, t, c_I, c_T). \tag{1}$$

When a predicted noise is calculated, Instruct-Pix2Pix follows the standard denoising process to update the image $z_t$ step by step and obtain the final result $z_0$, which is an edited image consistent with the initial image and text instruction.

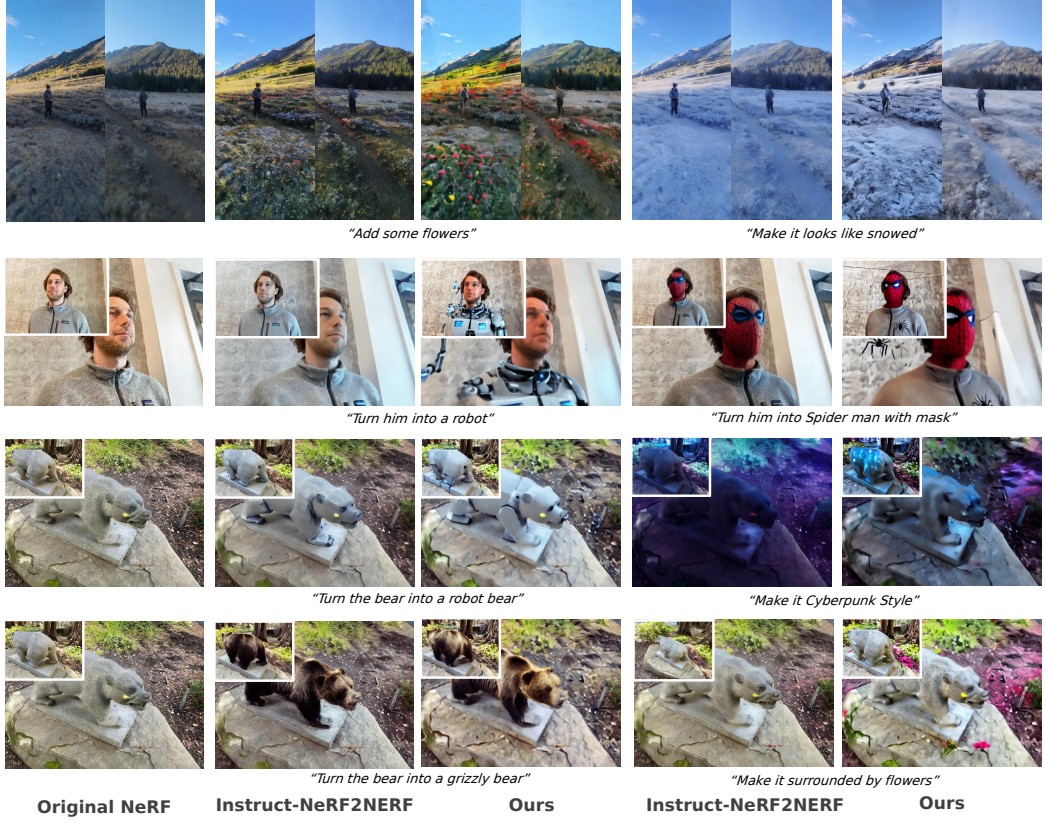

| Original NeRF | Instruct-NeRF2NERF | Ours | Instruct-NeRF2NERF | Ours |

Figure 4: **Qualitative comparison** with Instruct-NeRF2NeRF. Our ViCA-NeRF provides more details compared with Instruct-NeRF2NeRF. In addition, our ViCA-NeRF can handle challenging prompts, such as "Turn him into a robot," whereas Instruct-NeRF2NeRF fails in such cases.

## 3.2 Depth-Guided Sequential Editing

In our approach, we choose to reflect the user's 2D edits in 3D. Thus, we propose a sequential editing strategy. By defining certain key views, we sequentially edit them by projecting the previous edits to the next key view. This is the *geometric regularization* that explicitly enforces the consistency between different views.

**Sequential Editing with Mixup.** The overall procedure of editing is simple yet effective, which consists of editing key views and projecting to other views. First, we select a sequence of key views starting from a random view. Then, we sequentially use Instruct-Pix2Pix to edit each view and project the modification to the following views. At the same time, non-key views are also modified by projection. After editing all key views, the entire dataset is edited.

Given that we project several key views onto other views when they are edited, each view could be modified multiple times. Only pixels that have not yet been modified are altered to prevent inadvertent overwriting. Therefore, modifications from key views are mixed up with each view.

Consequently, our method yields a final result that aligns closely with the modified views, facilitating a *controllable* editing process. Since previously modified pixels are not changed until blending refinement (explained in Section 3.3), the edits of the first few views can dominate the final result.

**Key View Selection.** We employ key views to make the editing process consistent. This strategy allows the entire scene to be constrained by a handful of edited views. To ensure the edited views maintain consistency and bear similar content, we define the modified ratio $\rho$ for each view as the fraction of modified pixels. Then, we use the following equation to select the view with the largest

weight $w$ as the next key view:

$$w = \begin{cases} \rho & \rho < \phi \\ \phi - (\rho - \phi) & \rho \geq \phi, \end{cases} \quad (2)$$

where $\phi$ signifies the maximum desired ratio, which is set to $\phi = 0.3$ in all experiments. Intuitively, each time the next selected key view will have a suitable overlap with the edited region from each of the previous key views. The key view selection ends when all views' minimum $\rho$ exceeds a certain threshold.

**Depth-Guided Projection.** Modern NeRF models are capable of generating depth maps directly, thereby making it possible to construct geometric relationships between different views. We illustrate an example of projecting from edited view $j$ to view $i$ as below.

Given an RGB image $I_i$ and its corresponding estimated depth map $D_i$, the 3D point cloud associated with view $i$ can be calculated as:

$$p_i = P^{-1}(D_i, K_i, \xi_i). \quad (3)$$

Here, $K_i$ denotes the intrinsic parameters, $\xi_i$ represents the extrinsic parameters, and $P^{-1}$ is a function mapping 2D coordinates to a 3D point cloud.

To find the corresponding pixels on another view, we project this point cloud to view $j$ as follows:

$$I_{i,j} = P(I_j, K_j, \xi_j, p_i), \quad (4)$$

where $I_{i,j}$ signifies the color of the pixels projected from $I_j$ onto $I_i$. In practice, to ensure the correct correspondence and mitigate the effects of occlusion, we compute the reprojection error to filter out invalid pairs. By using this correspondence, we can project pixels from view $j$ to view $i$.

### 3.3 Blending Refinement Model

The depth information derived from NeRF [1] can often be quite noisy, resulting in noticeable outliers, so only applying the geometric regularization is insufficient. Specifically, while we can filter out incorrect correspondences, the remaining pixels and projections from varying lighting conditions may still introduce artifacts. Furthermore, a reasonable 2D edit in a single view may appear strange when considered within the overall 3D shape. To address these challenges, we introduce a blending refinement model that employs two Instruct-Pix2Pix [3] processes, acting as the *learned regularization* to improve the overall quality.

Details about the model are shown in Figure 3, where each diffusion process serves unique functions. During the first pass, we use the mixup image as input and the original image as a condition. This procedure aims to purify the noisy mixup image by preserving the structural integrity of the original image. The resulting image appears dissimilar to the consistent mixup image but without any noise. Thus during the second pass, we use the resulting image from the first pass as input and the mixup image as the condition. Given the shared semantic information between these two images, the outcome strives to closely align with the mixup image's detailed characteristics. Doing so allows us to swiftly generate a clean, stylized result, bypassing the need for iterative updates to the dataset during training.

An essential component of our refinement model involves using an average latent code for each of the two processes. We discovered that the diffusion model tends to generate diverse results, making it challenging to closely align with the target structure or content. However, this effect can be substantially mitigated by averaging multiple runs. Specifically, we introduce $n_r$ noised latent codes and update them independently. Finally, we compute the average of the latent codes and decode it back into the image. This method provides a more stable and consistent output, enhancing the overall quality of the editing process.

### 3.4 Warm-Up and Post-Refinement Strategies

To make the editing more efficient and further improve the quality, we propose a warm-up strategy along with a post-refinement strategy.

**Warm-Up.** When editing via instruct-Pix2Pix, the extent of editing changes can be modulated through hyperparameters, such as the guidance scale [15]. However, achieving the desired overall

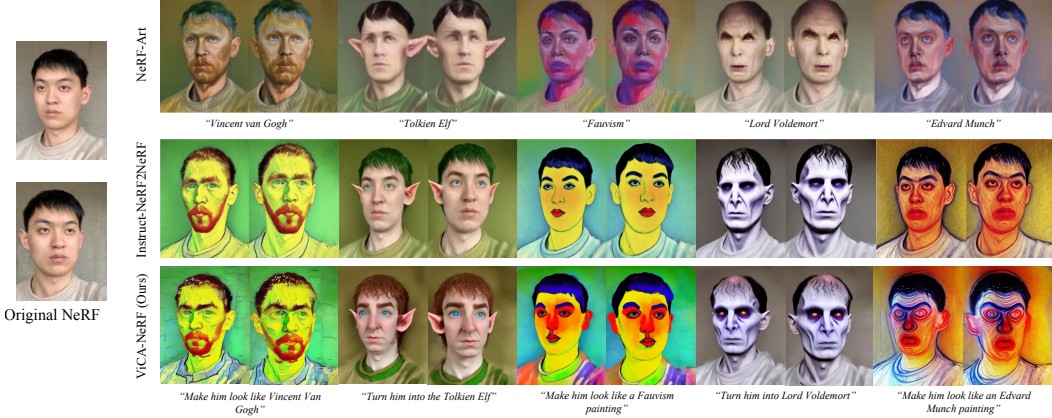

Figure 5: **Comparison on NeRF-Art.** We compare the editing results based on NeRF-Art's [34] sequences and edits. Our ViCA-NeRF produces more detailed information and achieves more substantial changes to the content.

style remains a problem. Although Instruct-NeRF2NeRF ensures consistent stylization, it necessitates iterative updates, which could inefficiently involve a large number of iterations.

To overcome these limitations, we introduce a strategy to efficiently warm up the editing process by blending edits directly. This approach is performed before the entire editing process. Specifically, we randomly select a view to edit, and then project the edited view onto all other views. For a given view $i$ and an edited view $j$, an update is calculated as follows:

$$I'_i = \lambda I_i \odot m_w + (1 - \lambda)I_{i,j} \odot m_w + I_i \odot (1 - m_w). \tag{5}$$

Here, $m_w$ denotes the binary mask for the projected area, and $\lambda$ is a hyperparameter employed to control the ratio preserved from the original pixel value. This efficient warm-up strategy accelerates the editing process while preserving a high degree of consistency in the stylization.

**Post-Refinement.** To further enhance the consistency of the edited results, we employ a post-refinement strategy once the NeRF has been trained on our updated data. We still utilize the modified Instruct-Pix2Pix with averaging architecture to generate a more accurate target for each view. Unlike the first stage, this time we employ the rendered image as input while continuing to use the mixup image as a condition. This post-refinement strategy further refines the consistency and quality of the final output, contributing to a more cohesive and visually appealing result.

## 4 Experiments

We conduct experiments on various scenes and text prompts. All our experiments are based on real scenes with NeRFStudio [9]. We first show some qualitative results and comparisons between our method and Instruct-NeRF2NeRF [2]. For artistic stylization, we test our method with the cases from NeRF-Art [34]. Experiments show that we achieve more detailed edits. Based on these scenes, we further conduct ablation studies on our method, including the effects of different components in the framework, the warm-up strategy, failure cases, and representative hyperparameters. We also conduct quantitative evaluation of the results, testing the textual alignment, consistency, and efficiency of our method.

### 4.1 Implementation Details

For a fair comparison, our method employs a configuration similar to that of Instruct-NeRF2NeRF. We utilize the 'nerfacto' model from NeRFStudio and Instruct-Pix2Pix [3] as our 2D editing model. For the diffusion model, we apply distinct hyperparameters for different components. During the editing of key views, the input timestep $t$ is set within the range [0.5, 0.9]. We employ 10 diffusion steps for this phase. In the blending refinement model, we set $t$ to 0.6 and $n_r$ to 5, and use only 3 diffusion steps. The reduced number of steps enables the model to operate more quickly, since encoding and decoding are executed only once. The diffusion model provides additional adjustable

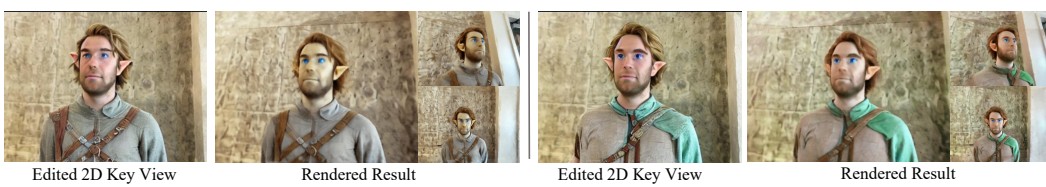

| Edited 2D Key View | Rendered Result | Edited 2D Key View | Rendered Result |

*"Turn him into Link from Zelda"*

Figure 6: **Different 2D edits and corresponding 3D edits.** Our ViCA-NeRF achieves high correspondences between 2D edits and 3D edits, thus enabling users to control the final edits. On the other hand, our method exhibits high diversity with different random seeds.

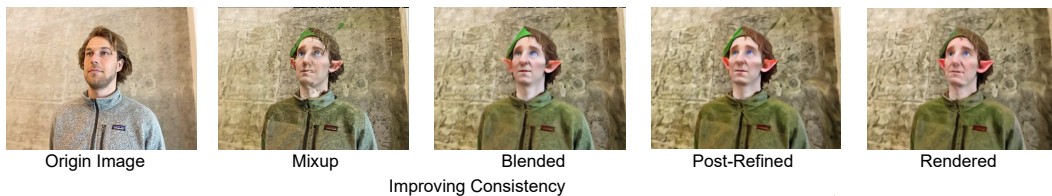

| Origin Image | Mixup | Blended | Post-Refined | Rendered |

Improving Consistency

Figure 7: **Image update through our ViCA-NeRF.** Our blending model effectively eliminates artifacts and boundaries caused by mixup. The post-refinement step further improves the consistency with the mixup image and the 3D content.

hyperparameters, such as the image guidance scale $S_I$ and the text guidance scale $S_T$. We adopt the default values of $S_I = 1.5$ and $S_T = 7.5$ from the model without manual adjustment.

For training the NeRF, we use an L1 and LPIPS (Learned Perceptual Image Patch Similarity) loss throughout the process. Initially, we pre-train the model for 30,000 iterations following the 'nerfacto' configuration. Subsequently, we continue to train the model using our proposed method. The optional post-refinement process occurs at 35,000 iterations. We obtain the final results at 40,000 iterations.

## 4.2 Qualitative Evaluation

Our qualitative results, illustrated in Figure 4 and Figure 5, demonstrate the effectiveness of our method in editing scenes through multiple views. We present results on three scenes with varying complexity, from human figures to large outdoor scenes. We also evaluate the stylization capabilities of NeRF-Art [34] and compare our results with the previous state of the art.

For comparison with Instruct-NeRF2NeRF, we replicate some cases from their work and assess several more challenging cases. For simpler tasks such as "turn the bear into a grizzly bear" or "make it look like it has snowed," our method achieves results similar to those of Instruct-NeRF2NeRF. This may be due to the inherent consistency of diffusion models in handling such scenarios. For more complex tasks that require additional detail, like transforming an object into a robot or adding flowers, our method shows significant improvements. A notable example is the "Spiderman cas," where Instruct-NeRF2NeRF struggles with differentiating lines on the mask, while our method renders a clear appearance.

When compared on NeRF-Art, our method clearly outperforms the previous techniques by exhibiting more details and adhering more closely to the desired style. Although we use the same 2D editing model as Instruct-NeRF2NeRF, we are able to produce more vivid colors and details, which are not achievable with iterative updates.

Moreover, a significant advantage of our method is that the final 3D edits can be directed through several key views. This allows us to achieve consistent edits not only across different 3D views but also between the 2D edits and the 3D renders, as shown in Figure 6.

## 4.3 Ablation Study

In our ablation study, we first evaluate the individual effects of mixup, blending, and post-refinement on the quality of editing. Further investigation includes the impact of warm-up iterations and NeRF

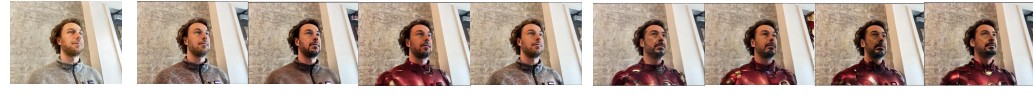

Origin Image                warmup=0                    warmup=10

Figure 8: **Edit distribution** with different warm-up configurations. When we use warm-up, the edit distribution becomes closer to the target distribution.

depth estimation accuracy. We also explore the influence of the parameter $\phi$ on key view selection for efficient editing. Each of these components is crucial for understanding the nuances of our method's performance.

**Model Component Ablation.** As shown in Figure 7, the mixup image may initially appear very noisy and contain artifacts. When we apply our blending model to refine it, the boundary from copy-pasting and artifacts is removed. However, while maintaining almost all the original content, the appearance of the image still changes slightly. Finally, when we apply our post-refinement strategy to update it, the result improves and becomes similar to the rendered result. Notice that the result after blending is sufficiently good, making further refinement an optional step.

**Warm-Up Iteration Ablation.** Considering that Instruct-Pix2Pix relies heavily on the input image, modifications to the input can significantly affect the output distribution. Therefore, to validate our warm-up procedure, which is designed to alter the scale of editing, we adjust this parameter from 0 (no warm-up) to 30. As Figure 8 illustrates, our findings indicate that while preserving the primary geometry, our warm-up procedure can effectively act as a controller for adjusting the scale of editing, introducing only minor computational costs.

**Ablation for Depth Estimation Errors.** The projection procedure is critical in our framework; therefore, inferior depth estimation may affect the final result. We conduct an ablation study focusing on two prevalent scenarios: glossy surfaces and sparser views. As Figure 9a demonstrates, our model can accurately make modifications to the bookshelf. If the reprojection check is omitted, more extensive changes are possible, albeit at the cost of losing original details. With sparser views, as depicted in Figure 9b, our model is capable of editing provided that the NeRF is successfully trained.

**Ablation on Key Views.** We perform experiments based on the hyperparameter $\phi$. Specifically, we test the results with $\phi$ set to 0, 0.3, and 1, as illustrated in Figure 10. When $\phi = 0$, the view with the least overlap is selected, resulting in a clear boundary (highlighted in the red box). In other cases, this issue is mitigated. However, when $\phi = 1$, nearly all views are designated as key views, which leads to reduced efficiency.

## 4.4 Local Editing

Existing methods often induce simultaneous changes in both background and foreground during editing, which contradicts specific user intentions that typically focus on altering distinct objects.

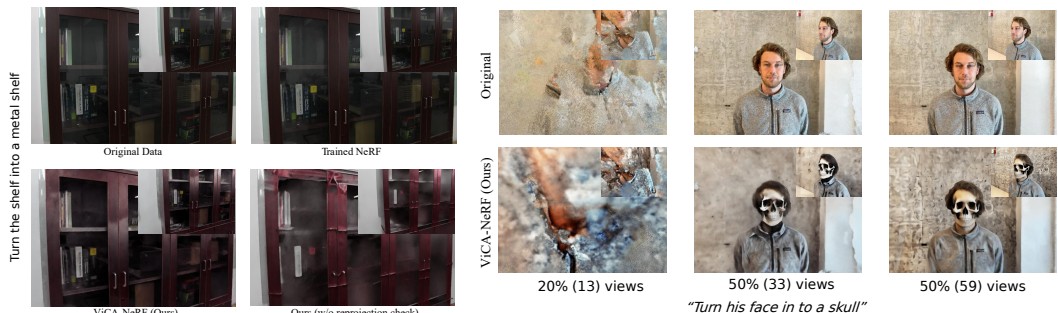

(a) **Results with glossy surface**          (b) **Ablation study on input views**

Figure 9: **Ablation study with incorrect depth.** We study both glossy surfaces and sparser views, where the results demonstrate that as long as NeRF is built successfully, our model can manage incorrect depth estimation.

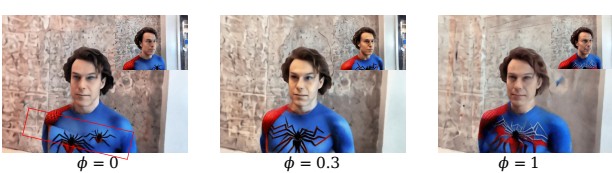

$\phi = 0$       $\phi = 0.3$      $\phi = 1$

*"Turn him into spider man"*

Figure 10: **Ablation study on $\phi$ in keyframe selection.** Our keyframe selection successfully avoids the boundary problem and achieves efficiency.

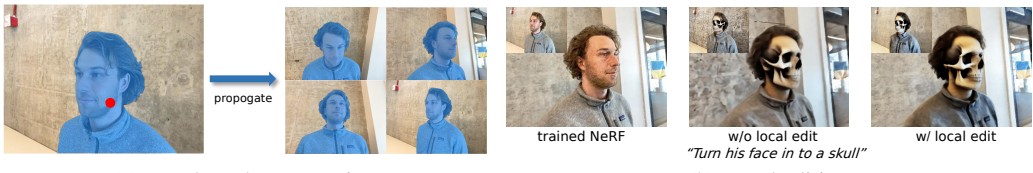

(a) Local mask propagation             (b) Local editing

Figure 11: **Application for local editing.** Our ViCA-NeRF can be applied with local editing. The edits can be done for the targeted object without changing the background (e.g., wall).

To address this discrepancy, we introduce a straightforward, user-friendly local editing technique within our framework. Initially, we employ Segment Anything Thing (SAM) [37] to derive masks encompassing all components within each image. Subsequently, users pinpoint the target instance with a single point to isolate it. Following this, we utilize the depth information to project the identified mask across different views progressively. This process continues as long as the projected mask sufficiently overlaps with the corresponding mask in the current view. Then, we integrate the wrapped segmentation mask with the segmented masks in the current view.

Figure 11 exemplifies the process, resulting in a multi-view consistent segmentation that permits exclusive modification of the human figure, leaving the background intact.

### 4.5 Discussion

For quantitative evaluation, we adopt the directional score metric utilized in Instruct-Pix2Pix [3] and the temporal consistency loss employed in Instruct-NeRF2NeRF [2]. Due to the undisclosed split and text prompts from Instruct-NeRF2NeRF, we establish various difficulty levels to assess performance. Results and comprehensive details are provided in the supplementary material.

A further benefit of our approach is the decoupling of the NeRF training from the dataset update process. This separation significantly reduces the computational burden typically imposed by the diffusion model at each iteration. The only computational expense arises from the warm-up, blending, and post-refinement stages. We analyze the time cost of the "Face" scene and compare it with Instruct-NeRF2NeRF on an RTX 4090 GPU. Both methods undergo 10,000 iterations; however, Instruct-NeRF2NeRF requires approximately 45 minutes, whereas our ViCA-NeRF only needs 15 minutes.

**Limitations.** Our method's efficacy is contingent upon the depth map accuracy derived from NeRF, which underscores our reliance on the quality of NeRF-generated results. Although our blending refinement can mitigate the impact of incorrect correspondences to some extent, the quality of the results deteriorates with inaccurate depth information. We also note that akin to Instruct-NeRF2NeRF, the edited outcomes tend to exhibit increased blurriness relative to the original NeRF. We investigate this phenomenon further and provide a detailed analysis in the supplementary material.

## 5 Conclusion

In this paper, we have proposed ViCA-NeRF, a view-consistency-aware 3D editing framework for text-guided NeRF editing. Given a text instruction, we can edit the NeRF with high efficiency. In addition to simple tasks like human stylization and weather changes, we support context-related operations such as "add some flowers" and editing highly detailed textures. Our method outperforms several baselines on a wide spectrum of scenes and text prompts. In the future, we will continue to improve the controllability and realism of 3D editing.

## Acknowledgements

This work was supported in part by NSF Grant #2106825, NIFA Award #2020-67021-32799, the Illinois-Insper Partnership, the Jump ARCHES endowment through the Health Care Engineering Systems Center, the National Center for Supercomputing Applications (NCSA) at the University of Illinois at Urbana-Champaign through the NCSA Fellows program, the IBM-Illinois Discovery Accelerator Institute, and the Amazon Research Award. This work used NVIDIA GPUs at NCSA Delta through allocations CIS220014 and CIS230012 from the Advanced Cyberinfrastructure Coordination Ecosystem: Services & Support (ACCESS) program, which is supported by NSF Grants #2138259, #2138286, #2138307, #2137603, and #2138296.

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
