# OpenReview forum: "ViCA-NeRF: View-Consistency-Aware 3D Editing of Neural Radiance Fields"
_NeurIPS.cc/2023/Conference — NeurIPS 2023 poster_

### Official Review · Reviewer_6ncD · 2023-06-27

**Soundness:** 4 excellent
**Presentation:** 2 fair
**Contribution:** 2 fair
**Rating:** 4
**Confidence:** 4

**Summary:**

This paper addresses the task of editing 3D scenes using text prompt. The method is based on performing 2D edits on different views using a diffusion model and then optimizing the 3D representation on the edited images. In particular, the authors propose two regularization methods to ensure the 3D consistency of the 2D edits: geometric regularization based on depth-aware warping of of the edited images to other views, and learned regularization by aligning the latent codes in the diffusion model. The proposed method is compared to two main baselines for editing of several 3D scenes.

**Strengths:**

-- Addressing the 3D consistency of text-based 3D editing is valuable, and is currently missing from the existing methods.

-- The proposed method has sufficient novelty containing different new components compared to the baseline.

-- The proposed regularization based on depth-aware warping is reasonable.

--  Not requiring the iterative editing and optimization used in the main baseline, Instruct-NeRF-to-NeRF (IN2N), enables a more efficient editing (3x faster based on what authors report).

-- The experiments show more promising results than the baseline in terms of 3D consistency.

**Weaknesses:**

-- I am mainly not convinced about the visual quality proposed methods, as the visual results seem to be quite blurry and lack details. Although IN2N also suffers from some blurry results, it seems the problem is intensified in the proposed method due to the averaging of the latent codes. I am not sure if the proposed averaging is a good solution, considering the blurry outcomes.

-- The method section is not well-written and it is quite hard to follow. I find some details of the proposed method to be either missing or not well-explained. Also, the diagrams visualizing the method are too general and not helpful enough in understanding the components of the proposed method. I am afraid the current representation of the paper might not be ready for publication.


**Questions:**

-- I did not fully understand how the key frames are selected, and how many key frames are used in the proposed method. I would appreciate if authors could further clarify that.

-- Are the key frames edited independently and then propagated to non-key frame views, or are the key frames also edited sequentially using the previously edited ones?

-- Is there any ordering when the edited key-frames warped to other views sequentially, or is the key frames are selected randomly and independently?

-- Some of the limitations of the method are discussed in the limitation section. It would have been more helpful to also include some visualizations of the failure cases, especially the ones which do not happen in IN2N.


**Limitations:**

The limitations and social impact have been addressed by the authors.

---

> ### Author Rebuttal · Authors · 2023-08-10
>
> We thank the reviewer for the valuable feedback. Below we address all the concerns.
>
> > Q1. it seems the burry problem is intensified in the proposed method due to the averaging of the latent codes
>
> A1. According to our experiments, the averaging process won’t make more blur results as shown in **Fig. R1** of the rebuttal PDF.  The main reason causing this blurry issue is the diffusion model. As our method wants to fully leverage the power of 2D editing,  As Instruct-Pix2Pix often changes borders and backgrounds erratically, the extensive 2D edits will result in artifacts for other views. When blending the image, though artifacts will be removed, the region will become smoother since some high-frequency information is unrecognizable. Thus, the result tends to have blurriness. Potentially, this can be improved from future 2D consistent diffusion models.
>
> For the question regarding "lack of details", we want to emphasize that we can actually make correct details for edits that are harder, e.g. spider man mask, Edward Munch painting, robot bear and other results shown.
>
> > Q2. How the keyframes are selected, and how many keyframes are used?
>
> A2. The number of keyframes depends on the scale of the scene, specifically, the range of camera movement. We sequentially sample keyframes until all views are covered by keyframes from a certain ratio. To sample the next keyframe, we calculate the covered ratio for each view and sort their weight to choose the frame with the largest weight. In practice, for the face scene with 60 views, 7 key views are sampled. However, the overall appearance is dominated by the first keyframe, e.g. **Figure I** in supplementary
>
> > Q3. Are the keyframes also edited sequentially using the previously edited ones?
>
> A3. Yes, they are sequentially edited. Though we allow automatic editing with text prompt, we also want to enable the user to guide the edits by key view editing, e.g. single view edits. As shown in Fig. I, we can control the whole edit with a single view.
>
> > Q4. Some visualizations of failure cases
>
> A4. Thank you for your suggestion. We include some hard cases in **Figs. R3 and R5** of the rebuttal PDF. Specifically, we showcase the results when the depth is worse, due to glossy surfaces or sparse view inputs. The glossy surface, e.g. glass, will cause incorrect depth. For sparse views, as the information of view consistency is reduced, the depth will become less accurate. While these challenges lead to worse performance, our approach can still make reasonable edits. Importantly, in contrast to previous methods like Instruct-NeRF2NeRF, which could experience editing failures in multiple scenarios, our approach is always able to edit the scene as long as the 2D diffusion model remains functional. However, the editing quality still depends on the quality of the NeRF representation. This also means that we did not encounter cases where our approach fails in scenarios that do not occur in Instruct-NeRF2NeRF.
>
> > Q5. Presentation of the paper: some details of the proposed method to be either missing or not well-explained
>
> A5. We hope that our detailed explanation and clarification provided above have addressed the reviewer’s concern. If any particular aspect remains unclear, we are more than happy to offer further elaboration in the discussion phase. Furthermore, we will improve the clarity of the revised paper, and will modify the methodology diagram to incorporate additional details for better understanding.

---

> > ### Author Response · Authors · 2023-08-10
> > **Additional clarification on certain blurry results**
> >
> > In addition to clarifying the reason behind certain blurry results as above, we would like to emphasize that **our model can work effectively in many cases where Instruct-NeRF2NeRF fails to converge**. For example, as shown in **Fig. 4** of the paper, our model is able to edit scenes under challenging instructions, as long as the 2D diffusion model remains functional. This is in contrast to previous methods like Instruct-NeRF2NeRF, which completely fail or produce insufficient edits in such situations. Our method also clearly demonstrates more details in some cases, e.g., as shown in **Fig. 5** of the paper. While our enhanced capability for broader editing and improved 3D consistency is notable, it does entail a trade-off by introducing noise/artifacts to the images, addressing which in our model causes blurriness in certain cases. However, based on the analysis presented earlier, we believe that using a diffusion model with improved controllability to 2D editing (e.g., only editing the desired objects/regions without changing the scene layout or other areas) would serve as a solution to mitigate the blurriness issue. We have validated this by showing an example of local editing in **Fig. R2** of the rebuttal PDF (using the Segment Anything Model to focus on the specific object to edit without changing the background). More generally, this blurriness issue can be mitigated by using an improved 2D diffusion model, which we leave as interesting future work.

---

> > ### Comment · Reviewer_6ncD · 2023-08-18
> > **Post-Rebuttal Comments**
> >
> > I appreciate the author's response to my questions. Here is my overall opinion about the work after reading the rebuttal and other reviews:
> >
> > The authors argue that the blurriness caused by the 2D diffusion model, and not the averaging component. However, looking at Fig R1 in the rebuttal PDF, it seems to me that the blurriness has clearly intensified compared to the blended image without averaging (E.G. the details of the hair, the ears, ...).
> >
> > In terms of the complexity of the method discussed by other reviewers, I also don't think the complexity on its own is a problem here, considering the difficulty of the task. However, one would expect the quality of the results to match the complexity of the method. Although the proposed pipeline is reasonable to me, I am still not convinced by the quality of the results obtained by the proposed method. I acknowledge the better 3D consistency of the method and its ability in converging in scenarios challenging for Instruct-Nerf2Nerf. However, such advantages are obtained at the cost of very blurry and unrealistic results.
> >
> > Regarding my concern about the quality of the presentation, the authors' response clarified the ambiguous parts for me. However, I believe the method section needs a significant amount of refining to be also clear for the readers, and I cannot defend the current version.
> >
> > Overall, I am reluctant to change my score for the same reasons as before: the quality of the results and the presentation of the paper are not convincing and might not meet the bar for NeurIPS. However, I also acknowledge the difficulty of the task and that the proposed method is reasonable and of enough novelty.

---

> > > ### Author Response · Authors · 2023-08-20
> > > **Clarification on remaining concerns (Part 1/2)**
> > >
> > > We thank the reviewer for recognizing the rationality and enough novelty of our work. We are glad that our response has clarified the ambiguous parts for the reviewer. We would like to provide further clarification regarding the remaining concerns raised by the reviewer.
> > >
> > > >Q1. It seems to me that the blurriness has clearly intensified compared to the blended image without averaging
> > >
> > > We would like to further clarify and explain why we believe it is the 2D diffusion model that caused some blurry results.
> > >
> > > Conceptually, two potential factors could contribute to the blurriness observed in 3D NeRF editing results: 1) inconsistent 2D edits from different views, so that when the NeRF aggregates the 2D edits from these different views, the resulting 3D edits become blurred; or 2) the loss of high-frequency information during 2D editing, which means that the 2D edits are already smoothed. In our case, because we have specifically improved the multi-view consistency of 2D edits across different views, the blurriness in 3D edits primarily stems from (2) the smoothed 2D edits, rather than (1) the inconsistent 2D edits. This is also evident in the fact that inconsistent edits tend to produce irregular texture on surfaces, but our results show much better regularity compared with the inconsistent edits from Instruct-NeRF2NeRF, e.g., Iron Man’s clothes shown in Fig. 1 of the main paper.
> > >
> > > Now the question is casted as investigating when the smoothing occurs in images, and we provide the corresponding analysis experiment in the rebuttal PDF. The result in Fig. R1 shows that the smoothing happens when the mixed image is passed through the diffusion model. We can see that in both cases (with or without averaging), most high-frequency information is lost, e.g., texture on the face, beard, and hair. As a result, the 3D edits via NeRF will be blurred. Particularly, the smoothing happens since the IP2P diffusion model tries to reduce the noise in the mixed image introduced by projection. Despite being provided the clear original image as guidance, the IP2P diffusion model fails to create very clear edits. In comparison, Instruct-NeRF2NeRF does not have this issue in its single update, since its input image does not contain such noise from projection. However, we would like to point out again that Instruct-NeRF2NeRF also faces the blurriness issue from iterative updates on the same view. Therefore, both Instruct-NeRF2NeRF and our method face the blurriness issue, but for different reasons.
> > >
> > > As for the difference between using and not using averaging, we would like to first clarify that **the results in Fig. R1 are intermediate image edits from the diffusion model, which cannot reflect the clarity in 3D after NeRF training**. In practice, though using averaging introduces minor blurriness to the 2D image, it actually reduces the blurriness and improves details **to the NeRF training**.  Particularly, the averaging greatly improves the consistency as it creates similar and more stable edits among different views. Please note that the example with Instruct-NeRF2NeRF in Fig. III of the supplementary uses a single pass from consistent images but results in very inconsistent edits (e.g., the clothes are changing rapidly for different views), and this is a similar scenario if we do not use averaging. We will include the comparison of the final rendered results from NeRF between using and not using averaging in the revision (which was not included in the rebuttal PDF due to limited space), where using averaging clearly shows better clarity.
> > >
> > > In summary, (1) both Instruct-NeRF2NeRF and our method face the blurriness issue, but for different reasons. (2) Some blurry results in our method are caused by the smoothing and loss of high-frequency information during 2D editing via the IP2P diffusion model.

---

> > > > ### Author Response · Authors · 2023-08-20
> > > > **Clarification on remaining concerns (Part 2/2)**
> > > >
> > > > > Q2. However, such advantages are obtained at the cost of very blurry and unrealistic results
> > > >
> > > > First, we respectfully disagree with the reviewer that our results are very blurry and unrealistic. Please note that **the blurriness happens for certain results, but not all**. A case in point is Fig. 5 in the main paper, where our results demonstrate better clarity, such as the wall and clothing for Vincent Van Gogh. Importantly, as shown in **Fig. R4** of the rebuttal PDF, the **user study involving 16 human raters** validates that our method produces significantly more realistic results than the state-of-the-art Instruct-NeRF2NeRF, with **substantially higher scores in user preferences**: our method achieved scores of 0.773 compared with 0.227 by Instruct-NeRF2NeRF for text faithfulness, and our method achieved scores of 0.875 compared with 0.125 by Instruct-NeRF2NeRF for diversity.
> > > >
> > > > Second, we would like to argue that for **3D editing tasks**, being able to achieve faithful 3D consistency should outweigh the occurrence of some blurriness. This is because ensuring 3D consistency represents a fundamental, imperative, yet a relatively unique challenge within 3D scenarios. However, it is under-explored in the field of NeRF-based 3D editing. Our work is the pioneering effort to explicitly address this issue and propose an effective strategy. In addition, as the reviewer recognized, our method is able to consistently converge and function even in challenging cases, in contrast to complete failure or insufficient edits as exhibited in Instruct-NeRF2NeRF. This property is desirable and very important in practical applications, where an editing system should always respond to the text instruction; an editing system that frequently fails to converge like Instruct-NeRF2NeRF makes it unusable in practice.
> > > >
> > > > Third, in our previous response, we have also suggested solutions to mitigating the blurriness issue. In particular, in **Fig. R2** of the rebuttal PDF, we have shown that by using SAM to generate a segmentation mask for the object of interest and then using our depth-guided propagation to create masks for all views, we successfully achieve local editing. Doing so reduces blurriness and greatly improves the clarity of background compared with previous methods. Note that this strategy cannot be accomplished by previous works, since it leverages depth-guided propagation which is our novel contribution.
> > > >
> > > > Finally, as mentioned in our previous response to Reviewer EWGc, by using the components from our method, **we offer an optional interactive editing interface for users** based on the selection of specific 2D editing. This enhancement significantly boosts the controllability and practicality of the editing process. We believe that such an interactive editing feature can be incorporated into upcoming research and will be advantageous for applications in this domain. Moreover, our method achieves significantly improved efficiency, with about 3 times faster than Instruct-NeRF2NeRF as highlighted by Reviewer EWGc.
> > > >
> > > > > Q3. I believe the method section needs a significant amount of refining to be also clear for the readers
> > > >
> > > > We greatly appreciate the reviewer’s suggestion on improving the quality of the presentation in the method section. We are committed to improving the presentation quality to the highest level in the revision. However, we respectfully disagree with the reviewer that the method section needs a significant amount of refining.
> > > >
> > > > First, we believe that most parts of our method section are clear and easy to read. As noted by Reviewer EWGc: “The paper is well-written and easy to follow. The figures are clear and most components of the method are adequately explained in the text.”
> > > >
> > > > In the meanwhile, we realize that some details of the proposed method are unclear to the reviewer. However, we believe that all these issues have been well explained in the rebuttal. As acknowledged by the reviewer, “the authors' response clarified the ambiguous parts for me.” Given that the detailed explanations have already been provided in the rebuttal and that these explanations have proven to be effective for clarifying confusion, we believe that integrating these explanations into the revision should be a straightforward task, requiring a relatively small amount of effort.
> > > >
> > > > Finally, we would like to express our sincere appreciation to the reviewer for the valuable comments, insightful suggestions, and considerable time and effort invested in evaluating our work. We hope that our clarifications provided in this response have addressed the reviewer's remaining concerns. Once again, thank you for your valuable feedback.

---

### Official Review · Reviewer_CQtY · 2023-07-06

**Soundness:** 3 good
**Presentation:** 2 fair
**Contribution:** 2 fair
**Rating:** 5
**Confidence:** 4

**Summary:**

The paper proposes a method to edit a NeRF using natural language. The authors achieve this by first “generating” an edited version of the input dataset and then training a NeRF on the edited dataset. To generate the said dataset, the authors train a NeRF on the original dataset for depth, with which they then propagate their keyframe edits (made with Instruct-Pix2Pix) to other views. Because NeRF depth is noisy, the authors use a diffusion-based blending network to refine those images into a cleaner dataset for the next-stage NeRF, which simply learns the radiance field from the new, edited images.


**Strengths:**

The paper tackles the difficult problem of editing 3D scenes in a multi-view-consistent way. The proposed method is general, being applicable to humans, objects, and even unbounded scenes. It also takes advantage of physics-based operations such as projections and combines them with learning to handle noise and uncertainty (e.g., in NeRF depth). The paper produces plausible results (only up to a low resolution though).

**Weaknesses:**


The major weakness, I think, is the complexity of the pipeline that requires the user to train a NeRF for depth, run inference on keyframes with Instruct Pix2pix, project edited frames to other views, run a diffusion model to perform blending, and finally train another NeRF on the edited images. It does not sound like an easy-to-use pipeline that can be flexibly extended or built upon. A complicated pipeline like this has many points of failure, too, as compared with a simpler pipeline.

The question then becomes which part of the pipeline we may cut to achieve similar results. The authors claim it’s crucial to perform diffusion-based blending to get around the artifacts due to direct mixup. Indeed, the results look quite blurry. Given (the 2nd-stage) NeRF also blurs the results, I think the authors should ablate the blending module, i.e., directly feeding the mixup output into the 2nd-stage NeRF. My hunch is NeRF’s blurring effect may already achieve what the additional diffusion model is doing, To be clear, I do think blurry results are a weakness of this paper, too.

I think one crucial experiment on the number of edited keyframes is missing: how many keyframes does one need to edit to achieve good results? Why not just run Instruct Pix2pix on all views instead of doing mixup and then blending?


**Questions:**

Because I don’t expect the authors to change their method, I’m primarily interested in hearing about points #2 and #3 from the Weaknesses section above.


**Limitations:**

Yes

---

> ### Author Rebuttal · Authors · 2023-08-10
>
> We thank the reviewer for the valuable feedback. Below we address all the concerns.
>
> > Q1. Complexity of the pipeline that requires the user to train a NeRF for depth, run inference on keyframes with Instruct Pix2pix, project edited frames…
>
> A1. We respectfully disagree with the reviewer that our pipeline is more complicated and less flexible in terms of extensibility compared with prior works like Instruct-NeRF2NeRF. Instead, as noted by Reviewer x1ev, “The motivation of the proposed pipeline is sound and makes sense. It is also simple, which makes it easy to follow.” Below we clarify the simplicity and extensibility of our pipeline from several important aspects.
>
> (1) **Our pipeline shares the same usage complexity with Instruct-NeRF2NeRF**. Much like Instruct-NeRF2NeRF, users are only required to provide a text prompt for editing. Our entire pipeline operates automatically, without necessitating manual intervention to sequence or  control its components.
>
> (2) **Our pipeline is designed in a highly structured, modular manner**, with each component contributing to improve the result while maintaining a degree of independence (as shown in Fig. 7). This design allows us to easily improve individual components separately, making the pipeline flexible and extensible.
>
> (3) Critically, different from the iterative update strategy used in Instruct-NeRF2NeRF, **our method decouples the processes of NeRF finetuning and data editing**, enabling direct acquisition of the final edited data. Thus, our method is more flexible to fit different NeRF models. In addition, by getting rid of iterative updates which cause about 6 times as long as the regular NeRF training, our method is much faster compared with Instruct-NeRF2NeRF.
>
> (4) Finally, we would like to emphasize that a pre-trained NeRF is always needed for the NeRF editing task. The depth is extracted from the intermediate density value and **does not need to be separately trained**. The mixup and blending process are designed to propagate sparse view edits to dense view edits.
>
> > Q2. Which part of the pipeline we may cut to achieve similar results?
>
> A2. As shown in Fig. 7 of the main paper and Fig. I of the supplementary, we systematically ablated different components of our method, and validated that all our components contribute to the improvement of editing results. More specifically, (1) regarding the mixup model, it propagates the key view edits, thus promoting global consistency. (2) Regarding the blending model, there will exist noticeable artifacts if it is removed, as ablated in Figs. 7 and I. This will be more significant for larger-scale scenes Indeed, as pointed out by reviewer CQtY, “Blending Refinement Model: Reduces noise in-depth information and resolves incongruities between 2D edits and their 3D context”. (3) In addition to these two modules, the warm-up process is designed to shift the image domain to target description efficiently, and the post-refinement further improves the results.
>
> As for the blurry results, it's mainly caused by the diffusion model. Different from Instruct-NeRF2NeRF where the blurriness is introduced by iterative updates, ours come from the artifacts. As shown in **Fig.R1**, when there are too many artifacts, some parts, e.g. the wall, will be smoothed. Thus, the final result will be blurred. We think this can be solved by a better 2D consistent diffusion model in the future. In addition, using the local editing strategy can directly get rid of the background-changing problem.
>
> > Q3. How many keyframes does one need to edit to achieve good results?
>
> A3. The number of keyframes depends on the scale of the scene, specifically, the range of camera movement. As each time we sample a keyframe until all the views are covered by a minimum ratio, more keyframes will be sampled on larger scenes. However, the whole editing process is automatic based on the input text prompt, similar to Instruct-NeRF2NeRF. Typically, for the face scene with 60 views, 7 key views will be taken.
>
> > Q4. Why not just run Instruct Pix2pix on all views instead of doing mixup and then blending?
>
> A4. Simply updating all views will lead to inconsistent edits and broken results. We did not discuss such a strategy, since it has been already ablated in previous work Instruct-NeRF2NeRF – Figure 8 “One time Update” in Instruct-NeRF2NeRF. Their observation is: “the initial edited 2D images are largely inconsistent, leading to blurry and artifact-filled 3D scenes.”

---

> > ### Author Response · Authors · 2023-08-10
> > **Additional clarification on certain blurry results**
> >
> > In addition to clarifying the reason behind certain blurry results as above, we would like to emphasize that **our model can work effectively in many cases where Instruct-NeRF2NeRF fails to converge**. For example, as shown in **Fig. 4** of the paper, our model is able to edit scenes under challenging instructions, as long as the 2D diffusion model remains functional. This is in contrast to previous methods like Instruct-NeRF2NeRF, which completely fail or produce insufficient edits in such situations. Our method also clearly demonstrates more details in some cases, e.g., as shown in **Fig. 5** of the paper. While our enhanced capability for broader editing and improved 3D consistency is notable, it does entail a trade-off by introducing noise/artifacts to the images, addressing which in our model causes blurriness in certain cases. However, based on the analysis presented earlier, we believe that using a diffusion model with improved controllability to 2D editing (e.g., only editing the desired objects/regions without changing the scene layout or other areas) would serve as a solution to mitigate the blurriness issue. We have validated this by showing an example of local editing in **Fig. R2** of the rebuttal PDF (using the Segment Anything Model to focus on the specific object to edit without changing the background). More generally, this blurriness issue can be mitigated by using an improved 2D diffusion model, which we leave as interesting future work.

---

> ### Comment · Reviewer_CQtY · 2023-08-13
> **Post-Rebuttal Update**
>
> I read the authors' rebuttal, which does address some of my concerns.
>
> Most importantly, I agree with Reviewer EWGc (and the authors) that putting together a system like this, albeit complex and difficult to build upon, has its own merits. The rebuttal explained in more detail how the different components of this pipeline are essential.
>
> As such, I am willing to improve my rating to BA given the task difficulty, the topic timeliness ("NLP x NeRF"), and the authors' rebuttal.

---

> > ### Author Response · Authors · 2023-08-14
> > **Follow-up on reviewer’s response**
> >
> > We wholeheartedly thank the reviewer for the encouraging response to our rebuttal. We are glad that the reviewer appreciates the merits of our work. We thank the reviewer for improving the rating.
> >
> > We also notice that there is some back and forth in the reviewer’s rating from weak accept to borderline accept. If the reviewer has any remaining concerns, please let us know. We are more than happy to address them.

---

### Official Review · Reviewer_iE4L · 2023-07-06

**Soundness:** 2 fair
**Presentation:** 3 good
**Contribution:** 2 fair
**Rating:** 6
**Confidence:** 3

**Summary:**

This paper introduced ViCA-NeRF, a method for 3D-consistent NeRF editing using diffusion models. The proposed method decouples NeRF editing with two steps. In first stage (i.e. dataset editing), key views are sampled and edited via Instruct-Pix2Pix model and propagated to other views using the NeRF depth maps, obtaining a mixup dataset. In second stage (NeRF training), the edited dataset can be directly used for training the NeRF model. The authors proposed  a blending refinement model to enhance the quality that enables consistent NeRF editing results.

**Strengths:**

The proposed method demonstrated 3D consistent NeRF editing results and often achieves better editing image quality than baseline Instruct-NeRF2NeRF and NeRF-Art. Concretely, here are some strengths of the proposed method:
- Guided Editing: The method uses key views for consistent and detailed 3D scene manipulation;
- Depth-guided Correspondence Matching: Allows projecting one edited view onto all others for consistency and detail;
- Key Frame Sampling and Mixup: This reduces the computational load by representing the entire scene with a few key frames;
- Blending Refinement Model: Reduces noise in depth information and resolves incongruities between 2D edits and their 3D context;
- Warm-Up and Post-Refinement Strategy: Enhances editing quality and efficiency;
- Semantically Meaningful Edits: The method can make significant edits based on textual prompts.

In summary, the method is capable of performing 3D consistent NeRF editing with text-guided diffusion pipeline. The quality of the results demonstrate improvements over previous approaches, sufficient ablation and comparisons are shown.


**Weaknesses:**

There are some limitations and concerns.
- Firstly, I was wondering what happens if the NeRF depth or geometry is bad, such as on glossy or specular surfaces. In such case, the depth-guided correspondence matching would decrease its accuracy. Considering most scenes shown in the paper are diffuse, it'd be interesting to have one such example as an ablation.
- Results are rather blurry, such as Flowers in Fig.1, Cyberpunk Style in Fig. 4 and Fireworks in Fig. VI. In contrast, the "original NeRF" on the left exhibits pretty sharp image details. What's the source of blurriness --- Is it due to key frame selection, depth quality, or diffusion model? Some analysis and (at least) discussions are necesssary and helpful.
- How's Equation (2) proposed? And the hyperparameter phi = 0.3.
- Can local edits be enabled? Currently the pipeline is, similar to Instruct-NeRF2NeRF, heavily based on Instruct-Pix2Pix. This is typically diffusion on the whole image (i.e. background is also changing and being modified). Would be nice to support locally editing of a person without touching the background in some cases.

**Questions:**

Some related works on 3D consistent NeRF editing:
- PaletteNeRF: Palette-based Appearance Editing of Neural Radiance Fields. CVPR 2023.
- Semantic-driven Image-based NeRF Editing with Prior-guided Editing Field. CVPR 2023.
- ARF: Artistic Radiance Fields. ECCV 2022.
- SNeRF: Stylized Neural Implicit Representations for 3D Scenes. SIGGRAPH 2022.
- NeRFshop: Interactive Editing of Neural Radiance Fields. I3D 2023.

**Limitations:**

Discussed in weakness section.

---

> ### Author Rebuttal · Authors · 2023-08-10
>
> We thank the reviewer for the valuable feedback. Below we address all the concerns.
>
> > Q1. What happens if the NeRF depth or geometry is bad, such as on glossy or specular surfaces
>
> A1. We investigate the robustness of our method against inaccuracy or failure in depth estimation in the rebuttal PDF. Per the reviewer’s request, we consider glossy surfaces, e.g. glass, which will cause incorrect depth. In this case, we are still able to propagate reasonable editing, as shown in **Figs. R3** of  the rebuttal PDF. Additionally, we evaluate the case with sparse view inputs, where the depth also becomes less accurate as the information of view consistency is reduced. **Figs. R5** of the rebuttal PDF shows that our method can still reach consistent edits as long as the NeRF is successfully trained.
>
> In the most extreme case where the estimated depth is completely wrong, there is no correct correspondence between different frames. Thus, the single-view editing won’t be able to propagate to other views. Our method will degenerate into a view-independent editing strategy, where each frame is updated through Instruct-Pix2Pix.
>
> > Q2. What's the source of blurriness
>
> A2. The blurriness mainly comes from the diffusion model. As the mixed image provide guidance for diffusion, if there are lots of artifacts after mixup, some high-frequency information is lost to match the target description. We show this phenomenon in rebuttal PDF Figure R1, where the wall tends to be kind of blurred. After the blending process, some regions are smoothed. For possible reason of artifacts, besides the inaccurate depth, the major problem is from Instruct-Pix2Pix. It tends to  Irregularly changes the object boundary or background, resulting in artifacts when projected. Regarding this, future diffusion models with better 2D consistency may help to reduce this problem.
>
> > Q3. How's Equation (2) proposed? And the hyperparameter phi = 0.3.
>
> A3. Equation (2) represents our keyframe selection criterion. Our intuition for Equation (2) is that each time we want the next selected keyframe to have a suitable overlap with observed regions from the previous keyframes. If the overlap is high every time, more keyframes will be sampled, which will affect the efficiency of our method. Conversely, if the overlap is low, the new edits will not take previous edits into account, resulting in inconsistent edits at the boundaries. Thus, we design a simple and straightforward piecewise linear function to assign the largest weight to frames that exhibit a certain overlap. The hyperparameter $\phi$ is chosen to balance efficiency and performance. When $\phi$ is large, frames with a higher overlap are encouraged, resulting in lower efficiency. When $\phi$is small, frames with a lower overlap are encouraged, which may cause inconsistent overlap regions. We show an ablation study on $\phi$ in **Fig.R6**
>
> > Q4. Can local edits be enabled?
>
> A4. Yes. Since our model can benefit from frame propagation, it is easy to integrate local edits by propagating the local mask automatically. Here we show one example by using Segment Anything Model (SAM) [Ref1] for local editing. The results are shown in **Fig. R2** of the rebuttal PDF. We can achieve local edits without touching the background.
>
> Specifically, we segment each image into regions by using SAM. Given a single view, the user indicates the object’s position, resulting in a segmentation mask. By propagating and mixing through depth, the highly overlapped regions in each frame are combined as the object mask. Then, only the masked regions are edited.
>
> > Q5: Some related works on 3D consistent NeRF editing
>
> A5: Thank you for your great suggestion! We will cite and discuss these papers in our revision. Below is a brief summary of comparisons between our work and these papers.
>
> PaletteNeRF: Rather than just changing the color, we manage to change the content according to text instructions.
>
> SINE: For real scenes, SINE focuses on changing the texture for a certain style or material, while our method focuses on general content editing.
>
> ARF: ARF tries to match the feature space to one reference image, where the main purpose is style changing.
>
> SNeRF: SNeRF uses an image stylization model and trains NeRF with the updated image. Instruct-NeRF2NeRF shares a similar flow as this work. By contrast, our method tried to directly modify a view-consistent dataset, which is much more efficient.
>
> NeRFshop: NeRFshop mainly deals with geometry editing of NeRF, while our method focuses on the content perspective.
>
> [Ref1] Kirillov, Alexander, et al. "Segment anything." arXiv preprint arXiv:2304.02643 (2023).

---

### Official Review · Reviewer_EWGc · 2023-07-06

**Soundness:** 3 good
**Presentation:** 3 good
**Contribution:** 3 good
**Rating:** 6
**Confidence:** 4

**Summary:**

This paper presents a method for editing Neural Radiance Fields using text instructions. Starting from an initial trained NeRF, the method samples a set of key views which are then edited using Instruct-Pix2Pix. Given the depth maps rendered from the NeRF, the method computes correspondences between different views and propagates the edits to the remaining views. To minimize the artifacts from warping the paper proposes to use a blending procedure that involves 2 Instruct-Pix2Pix models.

**Strengths:**

1. The paper identifies a known issue with performing per-frame edits of Neural Radiance Fields (inconsistencies between views) and proposes an intuitive method to address them. This involves selecting key frames and then use the NeRF geometry to propagate changes to the remaining views. The approach overall is technically sound.
2. Both the qualitative and quantitative results are good, and are on-par with Instruct-Nerf2Nerf. Given that the proposed method is about 3 times faster than Instruct-Nerf2Nerf then it is a valid, more lightweight alternative.
3. The paper is well-written and easy to follow. The figures are clear and most components of the method are adequately explained in the text.

**Weaknesses:**

1. The paper fails to show that all components of the method are needed. For example by looking at Figure 6, I cannot seem to see any difference between the "blended" and "post-refined" images.

**Questions:**

1. It is very hard to tell if the method is significantly better than Instruct-Nerf2Nerf. The quantitative evaluation in the Supplementary shows very similar numbers. The text-image direction score and the temporal consistency score are correlated with the perceptually better results but there is no direct relationship, so it's hard to quantify which method is better. At the same time, it's hard to tell from a handful of qualitative results which method is better. This is not a problem of this paper, it's generally an issue with editing papers where there is no "correct" solution. I do not have a good solution for this.
2. The keyframe selection criterion seems a little bit arbitrary. How does the value of $\phi$ affect the final result?

**Limitations:**

The authors have an extended discussion on the limitations of the method which I appreciate. There is also a discussion on the societal impact in the supplementary material.

---

> ### Author Rebuttal · Authors · 2023-08-10
>
> We thank the reviewer for the valuable feedback. Below we address all the concerns.
>
> > Q1. Cannot seem to see any difference between the "blended" and "post-refined" images in Fig. 6
>
> A1. The post-refinement process is designed to further improve consistency. While it may not make a significant difference, post-refinement can improve the consistency of 2D edits for the dataset. For example, in Fig. 7 (we assume the reviewer is referring to Fig. 7 instead of Fig. 6, as Fig. 7 provides the ablation study), after post-refinement,  the man’s hat becomes more consistent with the rendered result, displaying a shape and size that align mode closely with the rendered result. In Figure IV of the supplementary, after post-refinement, the reactor on Iron Man's suit exhibits enhanced and clearer details than before, with more distinct boundaries.
>
> > Q2. Hard to tell quantitatively if the method is significantly better than Instruct-Nerf2Nerf
>
> A2. We greatly appreciate the reviewer’s comment on the common evaluation issues in the field of NeRF-based 3D editing. While we followed the prior work on using the text-image direction score and the temporal consistency score as quantitative metrics, we agree with the reviewer that they are correlated with the perceptually better results but there is no direct relationship.
>
> To better quantify which method is better and clarify the reviewer’s concern, we further conducted a user study on the editing results and compared with Instruct-NeRF2NeRF, regarding the diversity of the editing results and how well they match the text description. The user study quantifies the comparison while being directly relevant to the perception. More detail is provided in the rebuttal PDF. And the result is shown in **Fig. R3** of the rebuttal PDF. **Our method achieves significantly higher preference in the study**.
>
> Moreover, we would like to emphasize that our main focus is to improve diversity, controllability, and consistency of 3D editing. Specifically, (1) as shown in Figs. I and II of the supplementary, as well as other qualitative results, we are able to achieve edits where Instruct-NeRF2NeRF fails and we also produce higher diversity. (2) Our method allows for user-guided edits from key-view edits or even single-view edits, which is more convenient. (3) Our method also provides efficient benefits, as noted by the reviewer.  (4) As suggested by reviewer iE4L, we further provide an application of local editing using our method in **Fig. R2** of the rebuttal PDF, which is infeasible for previous works.
>
> > Q3. Keyframe selection criterion seems a little bit arbitrary
>
> A3. Our design of the keyframe selection criterion is not arbitrary; instead, it follows the rationale to ensure that the next selected frame maintains a suitable overlap with the previous keyframes. Intuitively, if the overlap is high every time, more keyframes will be sampled, which will affect the efficiency of our method. Conversely, if the overlap is low, the new edits will not take previous edits into account, resulting in inconsistent edits at the boundaries. Based on this rationale, we use a simple and straightforward piecewise linear function for sequential keyframe selection, where the largest weight is assigned to frames that exhibit a certain overlap. Emplically, we found that our keyframe selection criterion works well in our experiments.
>
> > Q4. How does the value of $\phi$ affect the final result?
>
> A4. Per the reviewer’s request, we provide an ablation study on the hyperparameter $\phi$ in **Fig. R6** of the rebuttal PDF. When it's too large, e.g.1, the frame with the largest overlap will be chosen. Thus it will result in low efficiency, but the overall result will be consistent. When it's too small, the frame with the lowest overlap will be chosen. In such cases, the modification may follow different textures and color, making inconsistent edits on the boundaries of edits.

---

> ### Comment · Reviewer_EWGc · 2023-08-11
>
> I read the other reviews and the rebuttal. The rebuttal overall was positive, as the authors attempted to clarify the questions raised by the reviewers, and they provided additional quantitative and qualitative results.
>
> Currently the reviewers are split, with 2 giving WA and 3 BR.
> The main concerns and questions that the reviewers raised are:
> 1) The reliance of the method on possibly non-accurate depth estimates.
> 2) The not-so-clear keyframe selection strategy.
> 3) The overall complexity of the method
> 3) The evaluations.
>
> My overall take is that the method is not necessarily more complicated than other competing methods like Instruct-N2N. From the papers that I've read in the subfield of NeRF editing, it looks like there is a lot of engineering and tricks put into the methods to make them work, with limited "pure" research contributions. I believe however that such methods, if well put together still have merit and can be useful. I agree with Reviewer CQtY in their remark that probably it will be hard to build on top of this method because it's already fairly complex in terms of the total number of components stacked together.
>
> Regarding the user study, I would kindly ask the authors to provide more details about the protocol used to conduct it. In the supplementary material it says that the number of raters is 16, but the other details were missing. For example, how many examples were shown in total? How many renderings for each scene? Is it just the 10 scenes that were already used?
> I appreciate the fact that the authors did a user study, but if the number of examples is small, it's hard to draw statistically significant evidence. Obviously I want to mention here that the authors are not expected or required to perform a large scale study in the 1-week rebuttal timeframe.
>
> I would like to here the other reviewers' thoughts before committing to a final score.

---

> > ### Author Response · Authors · 2023-08-14
> > **Follow-up on reviewer’s response**
> >
> > We wholeheartedly thank the reviewer for the insightful and constructive comments. Below we address the reviewer’s follow-up questions.
> >
> > - more details about the user study
> >
> > Given that Instruct-NeRF2NeRF evaluates 6 scenes in their work, we conducted the user study on the same 6 scenes from Instruct-NeRF2NeRF for a fair comparison. The number of edits for each scene follows the size of their instruction pool. Specifically, we use 3 edits from the IN2N face, 2 edits from the bear, 2 edits from the NeRF-Art face, and 1 for the other 3 scenes (campsite, farm, and person), resulting in 10 samples. For each scene, the instructions are randomly sampled from the instructions shown in our work. With each instruction, we create a pair of edits with different random seeds, and each edit is rendered from 2 views. During the measurement, human raters are asked to determine which pair is more diverse and more in line with the description. The results are averaged over 16 human raters.
> >
> > We agree with the reviewer that our current user study might not represent a large-scale evaluation, primarily due to the time constraint during the rebuttal period and the lack of a large-scale benchmark with numerous scenes (which is a common issue in NeRF-based 3D editing research as explained in the previous response to Reviewer x1ev). On the other hand, we would like to kindly emphasize the **substantial disparities in user preferences** that emerged in the study: our method achieved scores of 0.773 compared with 0.227 by Instruct-NeRF2NeRF for text faithfulness, and our method achieved scores of 0.875 compared with 0.125 by Instruct-NeRF2NeRF for diversity. These results lead us to believe that our current user study already demonstrates a significant superiority of our method over Instruct-NeRF2NeRF. To provide a larger-scale evaluation, we are starting to undertake a more comprehensive user study, which features more scenes, samples, and raters, and we will include the results in the revision.
> >
> > - Complexity of the method
> >
> > We thank the reviewer for highlighting the intrinsic complexity of the task and the corresponding methodological complexity that is commonly found in the subfield of NeRF editing. Indeed, primarily due to the scarcity of large-scale annotated 3D datasets, 3D editing is challenging and needs to leverage different components like 3D modeling from NeRF and image editing from 2D diffusion models, address the associated gaps between 2D and 3D domains, and integrate them into a working pipeline.
> >
> > We also thank the reviewer for highlighting the merit and usefulness of our method. Here, we would like to further explain how subsequent works can build upon our method. (1) In the previous rebuttal (Response to Reviewer CQtY), we clarified that the structured, modular design of our pipeline facilitates its extensibility, so subsequent works can improve each component separately. (2) Our method brings insight into *explicitly* emphasizing 3D consistency during editing. This vital aspect has been overlooked in prior research and holds potential benefits for future studies. (3) By using the components from our method, **we offer an optional interactive editing interface for users** based on the selection of specific 2D editing. This enhancement significantly boosts the controllability and practicality of the editing process. We believe that such an interactive editing feature can be incorporated into upcoming research and will be advantageous for applications in this domain.

---

### Official Review · Reviewer_x1ev · 2023-07-19

**Soundness:** 3 good
**Presentation:** 3 good
**Contribution:** 2 fair
**Rating:** 4
**Confidence:** 4

**Summary:**

The paper proposes a method for 3D editing via text instruction that is view consistent. It leverages on the idea from Instruct-NeRF2NeRF, but additionally propagates the edits into multiple views in order to achieve multi-view consistency. To achieve such, a geometry regularization and a learned regularization were used, where the former uses depth for correspondence matching and the latter uses the typical 2D prior as used by Instruct-Pix2Pix but instead uses the average of the latent codes. Experiments show the effectiveness of the proposed method on 10 scenes.

**Strengths:**

The paper proposes a way to enable 3D editing through text instruction that is multi-view consistent. The motivation of the proposed pipeline is sound and makes sense. It is also simple, which makes it easy to follow. Experiment demos show nice qualitative examples as well as qualitative ablations.

**Weaknesses:**

The main concern I have with the paper is the thoroughness of the experiments. Specifically, in the main paper only qualitative results are included, and the scenes and examples presented are limited. The supplementary included a quantitative metric, but this is also only on 10 scenes.  It also mentioned that the pipeline is heavily reliant on depth, which makes one wonder whether the limited scenes shown is because of the failure cases with depth. Including failure cases might be helpful.

**Questions:**

A critical question I had is the details of the 10 training scenes, specifically, how many input views are used for training? The number of input views would give an idea on how critical the edit propagation to the different views is. Moreover, both the edit propagation and the geometric regularizer rely on depth inferred from the NeRF, and the correctness of NeRF depth is highly dependent on the number of training views. The failure case of the original Instruct-NeRF2NeRF is when the 2D diffusion samples is not consistent in 3D, which is also dependent on how many views are there. Perhaps an ablation on this would also be meaningful and make the contribution more convincing.

**Limitations:**

The authors included a limitation section in the supplementary material.

---

> ### Author Rebuttal · Authors · 2023-08-10
>
>
> We thank the reviewer for the valuable feedback. Below we address all the concerns.
>
> > Q1. Thoroughness of the experiments; the scenes and examples presented are limited
>
> A1. While we acknowledge the reviewer’s concern that the evaluated scenes might not be as numerous as one would expect, we would like to kindly point out that we have **evaluated more scenes and examples when compared with previous works**, such as Instruct-NeRF2NeRF [Ref1]. In fact, the lack of a large-scale benchmark is a common issue in NeRF-based 3D editing research, which is not specific to our paper. Constructing a benchmark with a large number of scenes is an important yet non-trivial work for future research, which is beyond the scope of this paper.
>
> While our focus is not on creating a large-scale benchmark, we did perform an experimental evaluation as thorough and comprehensive as possible. Specifically, (1) in addition to assessing all the scenes and examples presented in the SOTA Instruct-NeRF2NeRF, we extended our evaluation in the submission to include additional scenes such as the “person in the farm” scenario. (2) We validated our method **across a wide range of scenes**, from object-centric (e.g., “face”) scenes to in-the-wild unbounded (e.g., “campsite”) scenes. As noted by Reviewer CQtY, "the proposed method is general, being applicable to humans, objects, and even unbounded scenes.” (3) In **Fig. R7** of the rebuttal PDF, we have further expanded our evaluation to include extra scenes and examples. We adopt the existing NeRFStudio [Ref2] data and also transfer the data from NeRFRen [Ref3] with COLMAP. Again, this evaluation underscores that our method can be directly applied to various scenes and consistently achieve superior 3D editing results.
>
> Moreover, we would like to reiterate that we conducted both qualitative and quantitative evaluation in the submission. Due to the nature of the editing tasks, the ground truth for editing results is typically unavailable, partially making it difficult to conduct large-scale quantitative evaluation. Therefore, prior works like Instruct-NeRF2NeRF primarily focus on qualitative comparisons. As noted by Reviewer EWGc: “This is not a problem of this paper, it’s generally an issue with editing papers where there is no "correct" solution.” To further show the superiority of our method, we conduct a user study that evaluates both the diversity of editing results and how well they match the provided text description. We include the comparison with Instruct-NeRF2NeRF in **Fig. R4** of the rebuttal PDF. Our method achieves significantly higher preference in the study.
>
> > Q2. The pipeline is heavily reliant on depth and failure cases
>
> A2. First, we would like to clarify that while our method utilizes depth information inferred from NeRF, this does not indicate a significant dependence on the depth quality. Instead, our method is designed to be robust and not prone to failure when dealing with inaccurately estimated depth. This robustness is achieved through our blending model, which effectively removes the artifacts and guides the editing process towards a more consistent direction. Our empirical evaluation in Fig. 7 validates that our blending model successfully fixes the artifacts caused by incorrect depth and obtains reasonable geometry.
>
> We further investigate the robustness of our method against inaccuracy or failure in depth estimation in **Figs. R3 and R5** of the rebuttal PDF. We consider 2 types of cases: glossy surface and sparse view inputs. The glossy surface, e.g. glass, will cause incorrect depth. For sparse views, as the information of view consistency is reduced, the depth will become less accurate. **Notably, while the accuracy of depth will affect the final performance, we can still successfully edit the scenes as long as the NeRF is successfully trained**.
> Moreover, we would like to clarify that, as explained in A1, the scenes evaluated in this paper were not selected with the intention of avoiding depth-related failure cases. Rather, these scenes were assessed in accordance with prior works and to make a fair comparison.
>
> > Q3. Number of input views
>
> A3. Thank you for your suggestion. We are following the same settings as Instruct-NeRF2NeRF for a fair comparison. 90% of the images are used for training and 10% are used for validation. The number of input views varies, depending on the scenes. For small scenes, e.g. “face”, it contains 59 input views. For larger scenes, e.g., “person in the farm”, it contains 343 input views.
>
> Per the reviewer’s request, in **Fig. R5** of the rebuttal PDF, we have included an ablation study to investigate the impact of using a smaller number of input views, as this can result in worse depth estimation results. As we are using nerfacto [Ref2] which is not designed for dealing with sparse views, the training of NeRF will be broken when only 20% images are used (see Fig. R5). However, as long as the NeRF is successfully trained, even with less input views, our approach is still effective. By leveraging a more powerful base NeRF model that is designed to be able to tackle sparse views like Sparf [Ref4], our method is expected to work reliably with a small number of views.
>
> [Ref1] Ayaan Haque, et al. Instruct-NeRF2NeRF: Editing 3D scenes with instructions. arXiv, 2023.
>
> [Ref2] Matthew Tancik, et al. NeRFStudio: A modular framework for neural radiance field development. SIGGRAPH, 2023.
>
> [Ref3] Yuan-Chen Guo, et al. NeRFRen: Neural radiance fields with reflections. CVPR, 2022.
>
> [Ref4] Prune Truong, et al. Sparf: Neural radiance fields from sparse and noisy poses. CVPR, 2023.

---

> > ### Comment · Reviewer_x1ev · 2023-08-18
> >
> > Thank you for your detailed responses and for the additional experiments. I appreciate the effort put into the rebuttal. I fully agree that there is a lack of datasets and evaluation scheme in general for the task at hand. I also acknowledge that it is a difficult problem.

---

> > > ### Author Response · Authors · 2023-08-19
> > >
> > > Dear Reviewer,
> > >
> > > We wholeheartedly thank you for the encouraging response to our rebuttal. We are excited to know that all your concerns are well addressed. Considering your latest positive comment, we kindly request you to reconsider the rating of our paper, if possible. We would be more than happy to provide additional clarification if you have further questions. Thank you once again for your valuable feedback.

---

### Author Rebuttal · Authors · 2023-08-10

We thank all reviewers for their time and feedback. We are excited to read that Reviewer x1ev appreciated the quality of results and the rationality of the method; that Reviewer EWGc emphasized the motivation, efficiency of our method, and the clearness of the presentation; that Reviewer iE4L praised the validity and merits of different parts of our approach and provide inspiring related work; that Reviewer CQtY complimented that our method can work on various scenes; that Reviewer 6nCV underlined the novelty and promising results of our method. We address individual questions in replies to each review.

Based on your great suggestions, we made additional studies on our method. The results are shown in the PDF, and a brief summary is as follows:

* For the blurriness, we provide an explanation for such a phenomenon.

* We provide an application of local editing using our method with Segment Anything Model. Edits can be done for certain objects without changing the background.

* A user study regarding diversity and quality is provided.

* An ablation study on input views is conducted.

* We validate the effectiveness on glossy surfaces and noisy depth.

* Ablation studies on sparse input views.

* Additional editing results

---

### Decision · Program_Chairs · 2023-09-21

**Decision:**

Accept (poster)

**Comment:**

The reviewers appreciated the paper’s contribution of addressing the issue of inconsistency across different views when performing NeRF edits and its demonstrated improvement over prior baselines. Post rebuttal and discussion, there was some remaining concern regarding the blurriness of the results. Nonetheless, the overall contribution and improvement over prior work outweighs this concern. Please take into account all reviewer feedback in the camera-ready version.